# Turning a Novel Janus Electrospun Mat into an Amphiphilic Membrane with High Aromatic Hydrocarbon Adsorption Capacity

Nicolás Torasso [1,2], Paula González-Seligra [3], Federico Trupp [1,2], Diana Grondona [1,4] and Silvia Goyanes [1,2,*]

1 Universidad de Buenos Aires, Facultad de Ciencias Exactas y Naturales, Departamento de Física, Buenos Aires C1428, Argentina
2 CONICET-Universidad de Buenos Aires, Instituto de Física de Buenos Aires (IFIBA), Buenos Aires C1428, Argentina
3 Instituto de Ingenierías y Nuevas Tecnologías, CONICET-Universidad Nacional del Oeste, Buenos Aires B1718, Argentina
4 CONICET-Universidad de Buenos Aires, Instituto de Física del Plasma (INFIP), Buenos Aires C1428, Argentina
* Correspondence: goyanes@df.uba.ar; Tel.: +54-91167676667

**Abstract:** Aromatic hydrocarbons in water is one of the collateral effects of the petrochemical industry and represents a serious problem both for their toxicity and environmental contamination. In this work, an innovative amphiphilic membrane was developed capable of rapidly removing hydrocarbons (such as BTEX) present in water under the solubility limit. Firstly, a Janus nanostructured membrane was developed from the deposition of superhydrophobic carbonaceous nanoparticles (CNPs) synthesized by radiofrequency plasma polymerization on a hydrophilic electrospun poly(vinyl alcohol) mat. Secondly, this membrane was turned amphiphilic by UV exposure, allowing water to pass through. The surface properties of the membranes were studied through SEM, contact angle, and FTIR analysis. Dead-end experiments showed that the toluene and xylene selective sorption capacity reached the outstanding adsorption capacity of 647 mg/g and 666 mg/g, respectively, and that the membrane could be reused three times without efficiency loss. Furthermore, swelling of the PVA fibers prevented the liberation of NPs. The selective sorption capacity of the UV-exposed CNPs was explained by studying the interfacial energy relations between the materials at play. This work provides a simple, low-cost, and scalable technique to develop membranes with great potential for water remediation, including the removal of volatile organic compounds from produced water, as well as separating oil-in-water emulsions.

**Keywords:** carbonaceous nanoparticles; electrospinning; Janus; plasma polymerization; toluene; xylene



## 1. Introduction

One of the current technological challenges is the absence of efficient and economical materials capable of removing organic contaminants dissolved in water. A major source of organic contaminants in water is produced water (PW), i.e., wastewater generated during oil and gas production [1,2]. Studies estimate that the volume of produced water will exceed 605 million barrels per day by 2025, and its discharge directly into surface water or wells provokes high environmental impact and serious implications for the different ecosystems [1,3]. Currently, the scientific and industrial sectors are focusing on finding efficient and cost-effective treatment methods to reuse PW and achieve a zero-environmental harmful discharge. Among the most abundant groups of pollutants found in PW is the monocyclic aromatic BTEX group, which includes benzene, toluene, ethylbenzene,

and isomers of xylene (o-xylene, m-xylene, p-xylene) [3]. The concentration of BTEX in PW is typically 0.73–24.1 mg/L and 0.058–5.86 mg/L for toluene [4]. These pollutants have been designated as harmful to human health and the environment and are characterized by a very low water solubility (201 ppm, 520 ppm, 111 ppm, 130 ppm, 106 ppm, 111 ppm for benzene, toluene, ethylbenzene, o-xylene, m-xylene, p-xylene, respectively) [5,6].

Treatment of produced water is a rather complex process that involves a combination of physical, chemical, and biological methods to remove several contaminants [4]. Among physical treatments, membrane filtration is considered an efficient and advantageous process for treating PW [7]. One salient method of membrane fabrication is the electrospinning technique, which allows the mass production of porous membranes with selective wettability, high-specific surface area, and the advantage of being easily modified in different ways for specific purposes [8–10]. Poly(vinyl alcohol) (PVA) is one of the biodegradable polymers chosen to produce membranes by this technique because it is a low-cost, non-toxic, water-soluble polymer and has suitable properties that lead to its good electrospinnability [11,12]. Recently, PVA has been used for the development of electrospinning nanofibers with application in water remediation, achieving good performance [13–16]. It has been shown that when the size-exclusion filtration process is used, electrospun PVA membranes allow the successful retention of 20 nm nanoparticles with a rejection rate of up to 99% [17]. Furthermore, it is possible to synthesize biohybrid membranes from electrospun PVA mats and bacterial bioremediation agents capable of removing Cr(VI) and phenols [14]. However, in order to remove particles smaller than 20 nm or specific contaminants, it is necessary to add a capturing agent to the membrane [18].

Different carbon-based materials such as carbon nanotubes, activated carbon, graphene, graphene oxide, biochar, and carbonaceous nanoparticles have been extensively used as effective adsorbents for the removal of organic pollutants from contaminated water [5,19–23]. The adsorption process of volatile organic compounds (VOCs) on the carbonaceous nanomaterials can be explained in terms of different interaction mechanisms such as $\pi$-$\pi$ interactions, hydrogen bonding, and electrostatic interactions, depending on the functional groups present on the surface of the sorbent as well as the pH of the aqueous solution [24]. The pore size distribution and large, specific surface areas of carbonaceous sorbents increase the number of adsorption sites, and therefore, their adsorption capacity [5,25]. Yu et al. developed carbon nanotube composites and used the Langmuir and Freundlich adsorption models to analyze the adsorption process of toluene, xylene, and ethylbenzene. The authors found that the Langmuir model best fits the experimental data, obtaining maximum adsorption capacity values of 249, 227, 138, 106, and 63.3 mg/g for ethylbenzene, m-xylene, o-xylene, p-xylene, and toluene, respectively [26].

One effective technique for the development of carbonaceous sorbents capable of removing benzene and hexane from aqueous solutions is the RF plasma of acetylene [27,28]. Arias et al. showed that carbonaceous films from the polymerization of acetylene by radiofrequency (RF) plasma could interact with benzene through $\pi$ interactions, allowing the contaminant to be retained [28]. In a previous work of our group, Torasso et al. showed that it is possible to obtain plasma-synthesized carbonaceous nanoparticles (CNPs) with a size range of 8–200 nm and high oil sorption capacity [21]. Trupp et al. developed oil–water separation membranes by the deposition of the same CNPs on different substrates such as metallic meshes, polypropylene, nonwovens, and cotton fabrics. The hydrophobicity of the CNPs allowed oil flux through the membranes while preventing the passage of water [29]. All the used substrates had pores in the order of tens up to a hundred microns, which allowed high oil flux but also led to NPs leaching.

Electrospun PVA membranes could be promising carriers of hydrophobic carbonaceous nanoparticles developed by the RF plasma of acetylene. Considering the sizes of CNPs (nanoparticles larger than 20 nm) and according to Cimadoro et al., it is expected that once the PVA nanofibers are swollen, the nanoparticles will be trapped inside the membrane [17]. The use of hydrophobic nanoparticles is efficient in removing aromatic hydrocarbons. However, their hydrophobicity is a drawback when the material is to be

applied as a filter, where it is necessary to let water pass through the membrane. For this reason, it is necessary to modify the nanoparticles and, thus, the wettability of the membranes to achieve successful water permeation accompanied by effective hydrocarbon adsorption. Exposure to ultraviolet light is a simple and suitable approach to modifying and controlling the wetting properties of some materials [30]. Several authors have used UV light irradiation as a successful surface modification method for changing the wetting behavior of membranes [1,31]. Kusworo et al. showed that surface modification with UV irradiation leads to a decrease in the water contact angle of polyethersulfone membranes and provokes a water flux enhancement in these mats [1]. Particularly, it was found that by increasing the UV exposure dose, a hydrophobic plasma polymerized carbonaceous nanosponge becomes fully hydrophilic while keeping the oleophilic behavior [21].

In this work, we developed, by means of a simple, low-cost, and scalable technique, a nanostructured amphiphilic membrane capable of rapidly removing BTEX compounds present in water under the solubility limit by letting the water pass through while retaining the contaminant. The aim was to combine the high surface area, small-pore, hydrophilicity, and mechanical properties of electrospun PVA nanofibers, with the high hydrocarbon-sorption capacity of amphiphilic UV-exposed CNPs. The swelling of the PVA nanofibers solved the problem of the liberation of CNPs, increasing the synergy between the materials. This novel membrane shows great hydrocarbon-sorption capacity and is capable of removing toluene and xylene from water at high-efficiency rates. This technology has great potential for water remediation, including the removal of VOCs from produced water (such as BTEX), as well as separating oil-in-water emulsions due to their surface properties, morphology, and great sorption capacity.

## 2. Materials and Methods

### 2.1. Materials

Poly(vinyl alcohol), PVA (Mowiol 10–98) (Mw~61,000 g/mol), and citric acid (CA) were supplied by Sigma-Aldrich®. The hydrocarbons used in this work were commercial toluene, provided from Droquimar (≥99% purity), xylene from Biopack (mixture of isomers, ≥98.5% purity), and motor oil Elf Evolution 700 ST 10W40, purchased from a local store.

### 2.2. Preparation of Membranes through the Electrospinning Process

The PVA solution was obtained by dissolving 3 g of PVA and 0.15 g of CA in 25 mL of distilled water, followed by continuously stirring at 85 °C until complete dissolution. The PVA–CA solution was placed in a 20-mL syringe connected to a multi-nozzle injector with five needles, a feed rate of 0.4 mL/h per needle, and a constant tip-to-collector distance of 15 cm. The applied voltage was 30 kV. The nanofibers were collected using a grounded rotating drum of 6 cm diameter covered by aluminum foil. An electrospinning equipment scheme is shown in Figure 1a. Electrospun mats were treated at 190 °C for 10 min to achieve water stability by crosslinking the PVA with CA [32].

### 2.3. Plasma Polymerization and UV Irradiation

Carbonaceous nanoparticles were deposited by an RF glow discharge on electrospun membranes. The plasma deposition process was similar to that reported by Trupp et al., except that in the present work, the smaller nanofiber diameter produces a more elastic material, allowing the nanoparticles to be inserted into the membrane. CNP deposition was implemented using a reactor consisting of a vacuum chamber made of stainless steel with immersed cylindrical concentric electrodes separated from each other at a distance of 4 cm, as shown in Figure 1b. The internal electrode was connected to the RF power supply (13.56 MHz, 600 W, RF VII Inc., Newfield, USA) through a matching unit, and the other electrode was grounded. The chamber was evacuated to a base pressure of 0.4 mbar using a rotary vane pump and then filled with acetylene ($C_2H_2$) of commercial grade up to a working pressure of 4.5 mbar. The discharge duration was 4 min with an applied output RF power of 40 W.

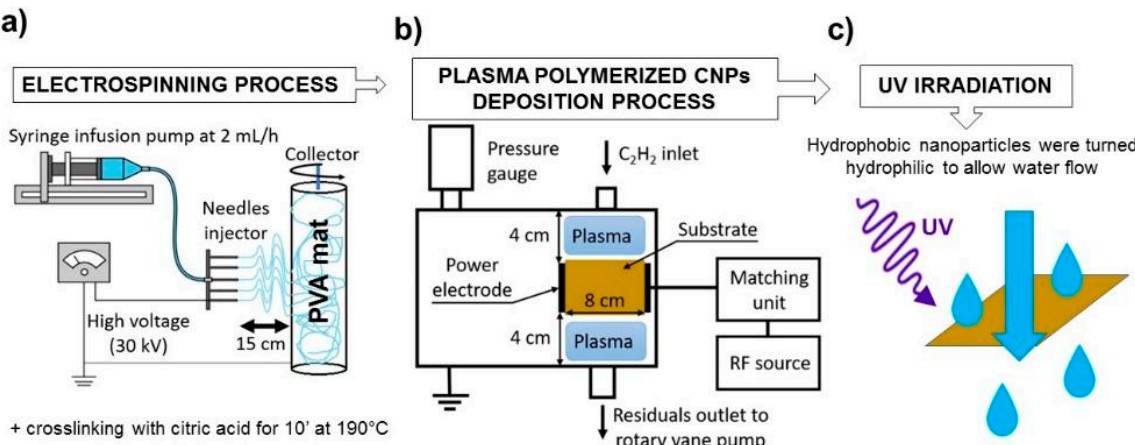

**Figure 1.** Schematic representation of the manufacturing process of the amphiphilic membranes. (**a**) PVA electrospinning process. (**b**) Plasma polymerized CNPs deposition on PVA membrane (**c**) UV exposure of PVA-CNP membrane.

In order to change their wettability, the membranes were irradiated using a UV lamp (Philips HPA 400 s, 400 W) at 75 cm from the sample for 120 min (irradiation power of 42 W/m$^2$ and total dose of 302 kJ/m$^2$).

Figure 1 shows the scheme of the complete process to obtain the final amphiphilic membranes consisting of the electrospun PVA mat combined with the plasma polymerized carbonaceous nanoparticles and their posterior modification with UV radiation. We denominate Side A and Side B of the membrane as those corresponding to PVA nanofibers and CNPs, respectively.

*2.4. Characterization*

2.4.1. Morphological Characterization

The surface morphology of the membranes was characterized by Field Emission Scanning Electron Microscopy (FE-SEM Zeiss LEO 982 GEMINI). Samples were sputter-coated with a thin layer of platinum (15 s, 0.06 mbar of Ar) before being tested. The average diameter of nanofibers and CNPs was determined from different SEM images using ImageJ software [33]. The size distributions were presented as histograms.

2.4.2. Contact Angle and Surface Tension

The wetting properties of membranes were evaluated from contact angle measurements. Drops of 13 µL of distilled water, toluene, and motor oil Elf Evolution 700 ST 10W40 were deposited on the membranes at an ambient temperature. Contact angles were measured after 6 s using an Optical Tensiometer (One Attension Theta-Biolin Scientific, Gothenburg, Sweden). In order to study the change in membrane hydrophobicity caused by UV exposure, the contact angle of the membranes was evaluated before and after UV irradiation. The reported values of this parameter are the average of four measurements for each sample.

Toluene-Water interfacial energy ($\gamma_{T\text{-}W}$) was measured in the laboratory using the same optical tensiometer.

2.4.3. Adsorption Capacity

Filtration experiments were performed using a dead-end apparatus consisting of a filter holder and an infusion pump injecting 5 mL of either toluene or xylene in distilled water (40 mg/L) at 10 L h$^{-1}$ m$^{-2}$ (Figure 2). The membrane was positioned such that Side B was facing upwards and in contact with contaminated water. This procedure was repeated seven times, giving seven filtration cycles. The whole apparatus was closed to avoid contaminant evaporation during the experiment. Toluene and xylene concentrations were

measured by UV-Vis spectroscopy (Shimadzu UV-1800 spectrophotometer in absorbance mode) immediately after filtration using the maximum absorption at 206 nm, corresponding to the main characteristic peak of both toluene and xylene. The filtration efficiency (FE) was calculated according to the following equation:

$$FE = \frac{C_0 - C}{C_0} \times 100\% \tag{1}$$

where $C_0$ and $C$ are the concentration before and after filtration, respectively. Adsorption capacity ($q$) was calculated as:

$$q = \frac{m_{tol/xyl}}{m_{CNP}} \tag{2}$$

where $m_{tol/xyl}$ is the total mass of either toluene or xylene removed from water and $m_{CNP}$ is the total mass of CNPs in the membrane.

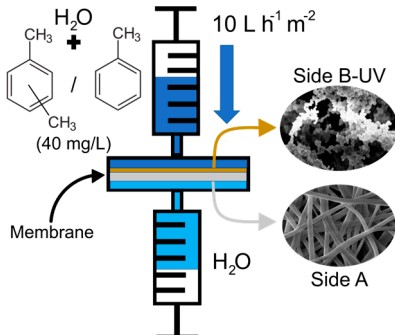

**Figure 2.** Schematic representation of the dead-end filtration apparatus used for the toluene adsorption experiment.

A control experiment to assess the PVA adsorption of toluene was performed. An in-put solution of 10 mL having an initial toluene concentration of 35 mg/L was filtered through a PVA membrane without CNPs at 10 L h$^{-1}$ m$^{-2}$.

### 2.4.4. Fourier-Transform Infrared Spectroscopy with Attenuated Total Reflectance (ATR-FTIR)

The surface chemistry of both sides of the membrane and the effect of UV exposure and toluene adsorption were evaluated by ATR-FTIR spectroscopy. The infrared spectra were recorded on a Jasco FTIR 4100 spectrophotometer using the Attenuated Total Reflectance (ATR) mode with a ZnSe crystal instead of the Transmittance mode to evaluate the differences between both sides of the membrane. For each sample, 64 scans were recorded in the range 4000–600 cm$^{-1}$ with a resolution of 2 cm$^{-1}$ and incidence angle of 45°.

## 3. Results and Discussion

### 3.1. Membrane Morphology

Plasma treatment on one side of the PVA electrospun membrane led to a Janus-like material with different chemical and morphological properties on each side (as shown by the wettability demonstration in Video S1). Figure 3 shows photographic images of both sides of the PVA-CNP Janus membrane. Side A shows the untreated side of the PVA membrane with the characteristic white appearance of electrospun PVA. Side B shows the treated side of the membrane in which plasma polymerization on the surface can be appreciated as a uniformly yellow-orange coating over the PVA membrane. The plasma polymerization optimized process allows a homogeneous coverage of the membrane surface. The thickness of the PVA electrospun mat was (140 ± 6) μm, while that of the full membrane (after plasma treatment) was (175 ± 7) μm. Therefore, the CNP layer thickness is (35 ± 9) μm.

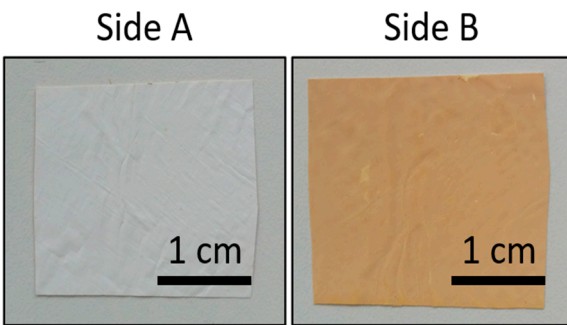

**Figure 3.** Photographs of the PVA-CNP Janus membrane. Side A: uncovered. Side B: coated with plasma polymerized CNPs.

### 3.2. SEM Analysis

The unique morphology of each side of the PVA-CNP Janus membrane was studied by FE-SEM and is shown in Figure 4a–d. Side A presented the typical randomly oriented continuous nanofibers of crosslinked PVA mats with an average diameter of $(160 \pm 30)$ nm (Figure 4e). It has a mean surface density of $(1.2 \pm 0.1)$ mg$_{PVA}$/cm$^2$. The observed bead-free morphology and uniformity of the fibers can be attributed to the optimized electrospinning conditions. Besides, water insolubility and structural integrity of nanofibers were achieved by crosslinking with citric acid, as reported in previous works [34,35]. The observable pores between the fibers allow water flux with low-pressure drop compared to nano or ultrafiltration membranes [17].

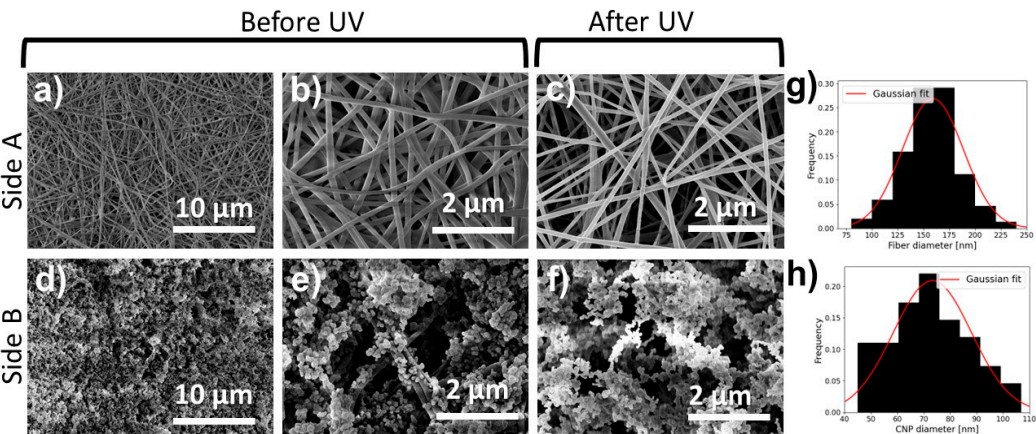

**Figure 4.** FE-SEM images of the PVA-CNP Janus membrane on Side A (**a**,**b**) and Side B (**d**,**e**) and of the PVA-CNP membrane after UV irradiation on Side A (**c**) and Side B (**f**). Histograms of both the nanofibers (**g**) and the CNPs (**h**) diameter distribution are shown on the right.

Side B was exposed to the acetylene-rich plasma phase, as depicted in Section 2.3. PVA nanofibers were covered with CNPs, which were found to have similar characteristics to those deposited on polypropylene, nonwoven, and cotton fabric membranes [29]. As seen in Figure 4d,e, CNP coating is composed of an agglomeration of individual nanoparticles with diameters ranging between 45 and 105 nm (Figure 4h), providing the membrane with a hierarchical porous micro-nanostructure that enhances the surface-to-volume ratio. These CNPs are homogeneously distributed along the surface, with a surface mass density of 300 µg/cm$^2$. Figure 4c,f show Side A and Side B of the membrane after UV exposure. As can be observed in Figure 4f, the morphology and sizes of the nanoparticles were not affected by the UV treatment.

### 3.3. FTIR Studies

Apart from differences in morphology, each side of the membrane has a significantly different chemical structure, as depicted by the FTIR spectra (Figure 5). Side A presents the

characteristic bands of PVA. For example, those associated with polar groups stretching vibrations O-H (at 3300 cm$^{-1}$) and C-O (at 1088 cm$^{-1}$). The bands at 2939 cm$^{-1}$ and 2909 cm$^{-1}$ are related to symmetric and asymmetric stretching vibrations of the C-H bond in CH$_2$. The band at 1430 cm$^{-1}$ corresponds to bending vibrations of C-H in the CH$_2$ group, and that at 845 cm$^{-1}$ to C-C stretching vibration [15,36].

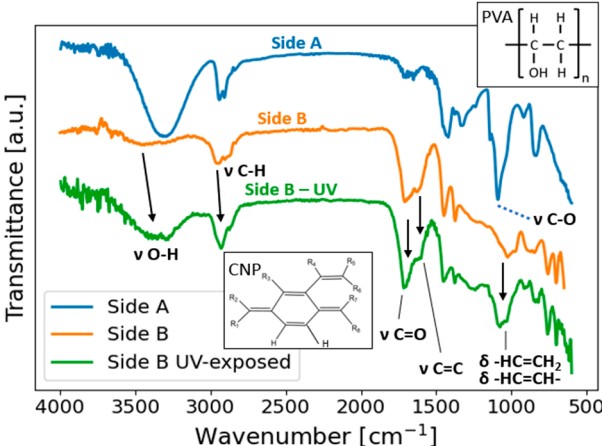

**Figure 5.** FTIR spectra of the Janus membrane: Side A (blue), Side B (orange), and Side B after UV exposure (green). The inset shows the predominant chemical composition for each case.

On the contrary, Side B chemistry is governed by the superhydrophobic CNP structure, displaying the vibration bands associated with conjugated C=C in the carbonaceous structure near 1620 cm$^{-1}$ and symmetric and asymmetric stretching vibration of C-H between 2850 cm$^{-1}$ and 3000 cm$^{-1}$. Structures like -CH=CH$_2$, -CH=CH- *cis* and *trans*, and double substituted alkenes also show absorption peaks between 1600 cm$^{-1}$ and 1700 cm$^{-1}$, as suggested by Torasso et al. [21].

Changes in the chemical composition of Side B after UV exposure become evident in FTIR spectra (Figure 5). These changes are associated with polymer oxidation and the formation of carbonyl groups such as carboxylic acid [37], which is suggested by the significant increase in the O-H vibration band relative to C-H stretching near 3000 cm$^{-1}$ and an increase in symmetric stretching vibration of C-O around 1220 cm$^{-1}$. Furthermore, results suggest that UV exposure reduced the number of unsaturations of the CNP, leading to a decrease in C=C stretching at 1615 cm$^{-1}$ and an increase and shift of C-H stretching band from 2955 cm$^{-1}$ to 2929 cm$^{-1}$. Similar changes were previously reported on UV-exposed CNPs deposited over Al substrates [21].

The calculation of the penetration depth of IR irradiation in a continuous polymeric medium with the specifications of this work results in the range between 0.5 μm and 6 μm depending on the wavelength and refractive index of the material (1.48 for PVA and typically from 1.52 to less than 1.64 for plasma polymerized films using acetylene [38,39]). These numbers may also be reduced by the fact that both sides of the membrane present porous structures. Considering the thickness of both layers of the membrane (PVA fibers and CNPs) reported in Section 3.1, it is possible to conclude that the chemical information shown by the FTIR spectra corresponds to each respective side.

### 3.4. Membrane Wettability

Figure 6 shows the contact angles of water ($\theta_W$), toluene ($\theta_T$), and oil ($\theta_O$) of Side A, Side B, and Side B after UV treatment. While Side A is a hydrophilic surface displaying $\theta_W = (43 \pm 3)°$, Side B of the membrane presents a superhydrophobic character with a contact angle $\theta_W = (157 \pm 3)°$. Both sides of the membrane present great wettability to toluene and oil: $\theta_T \cong 0°$ and $\theta_O \lesssim 10°$. The wetting behavior of the membranes depends on the chemical composition and morphology of the surface [40]. In this case, when a drop of water is placed on Side A, the affinity between PVA and water leads the droplet to wet

the surface, and the rugosity and porosity of the fibers reduce the apparent contact angle. On the other side, the chemistry of the CNPs leads to reduced wetting, which is further hindered by the air trapped in the micro-nanostructure, forming the Cassie–Baxter state.

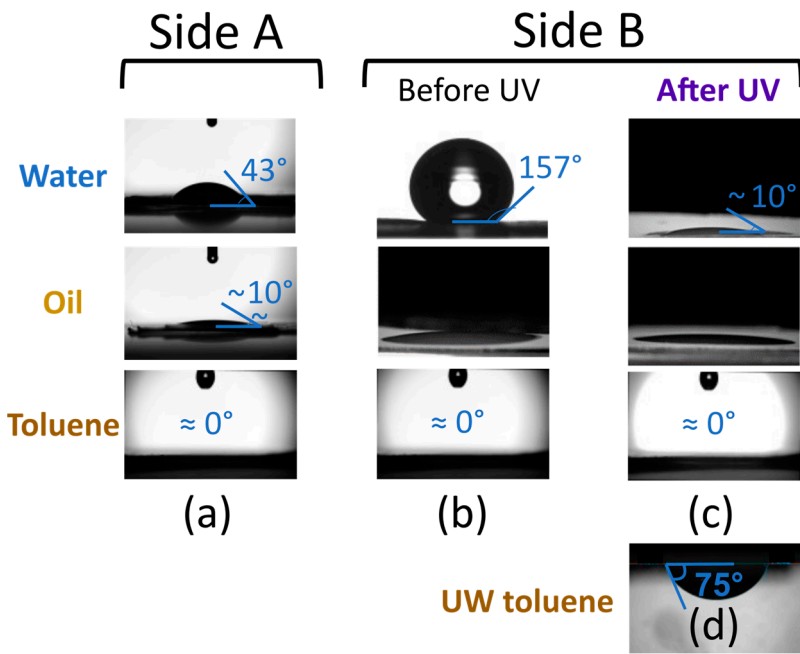

**Figure 6.** Water, toluene, and oil contact angle on Side A (**a**), Side B (**b**), and Side B after UV treatment (**c**). The underwater (UW) toluene contact angle on Side B after UV treatment is shown in (**d**).

Superhydrophobic materials have enormous potential as selective filters or adsorbents for low surface energy water contaminants [21,29]. In particular, the PVA-CNP Janus membrane developed herein is a great candidate for water remediation because of its high surface area, and superhydrophobic and oleophilic nature, as demonstrated by water and oil contact angle measurements (Figure 6). In spite of these features, when hydrocarbons are present in low concentrations such that no phase separation is observed (like the case of BTEX below the solubility threshold), the superhydrophobic nature of Side B prevents contaminated water from entering the nano-microstructure of the material and thus severely limits the chance of absorbing the contaminant. In order to overcome this limitation, we proposed the surface modification of the CNPs by UV irradiation. This methodology has been tested in a previous work of the group to assess the effect of sunlight on the chemical properties of CNP powder, and it was shown that UV exposure leads to an increase in carbonyl and hydroxyl groups, changing the wetting properties of the material [21]. Figure 6 shows that a proper UV treatment can drastically change the water contact angle of the PVA-CNP membrane ($\theta_W \lesssim 10°$) by providing the nano-microstructure with hydrophilic sites necessary for water diffusion in the porous network. The surface chemistry changes of the membrane were confirmed by the presence of hydroxyl and carbonyl groups in the FTIR spectrum (see Figure 5). The new chemical conformation of the membrane allows water to spread and leads to the full wetting Wenzel state necessary for toluene adsorption from water.

When toluene or oil was dropped on the surface of Side B, the droplet was immediately absorbed independently of UV exposure (Figure 6). The underwater toluene contact angle on Side B after UV treatment was 75°, as shown in Figure 6d, evidencing the consistent oleophilic character of the carbonaceous nanoparticles. Therefore, the Janus membrane turned amphiphilic after UV exposure, that is, with hydrophilic and superoleophilic properties ($\theta_W \lesssim 10°$, $\theta_T \cong 0$, $\theta_O \lesssim 10°$). The amphiphilic character of membranes allows the separation of oil-in-water emulsions, as oil is retained in the membrane while water is able

to immediately permeate through [40]. This particular property of the membrane offers a potential application as a filter material with hydrocarbon adsorption capacity.

### 3.5. Toluene Adsorption

The incorporation of polar groups in the nanostructure of the UV-exposed CNPs (UVCNPs) resulted in a hydrophilic material, as shown in CA measurements, and thus allowed their use as water filtration membranes. The ability of the PVA-UVCNP membrane to absorb toluene and xylene was evaluated using UV-VIS spectrophotometry (Figure 7a). The initial solution (before permeation) presented a broad band in the range of 200–220 nm corresponding to toluene/xylene absorption [41]. The intensity of this band decreases significantly for permeated solution curves. As indicated in the experimental section, in order to avoid the evaporation of toluene, the absorption was performed in a closed system, and the absorbance was measured immediately after sampling. Therefore, the decrement in the intensity of the band in the range of 200–220 nm demonstrates that toluene was retained by the PVA-UVCNP membrane. An analogous permeation study through a control PVA electrospun mat without the CNPs showed that PVA did not absorb toluene. This is shown in the UV-Vis spectra of the solution before and after permeation through the PVA electrospun mat (Figure S1).

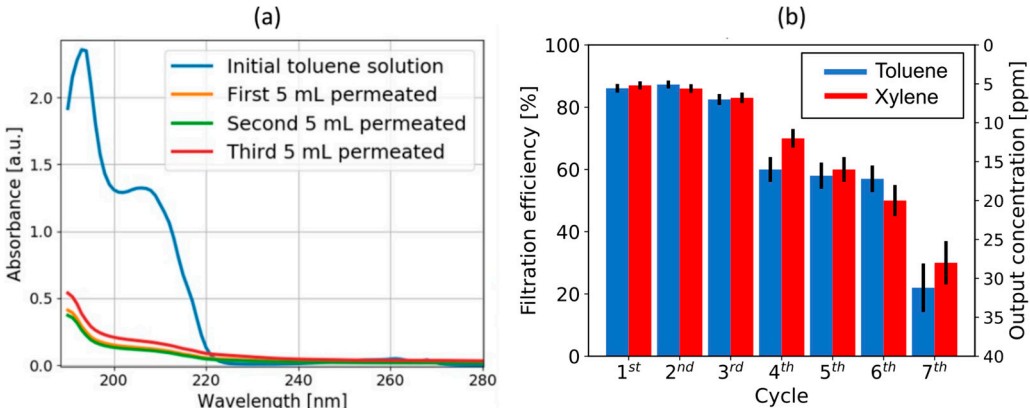

**Figure 7.** (**a**) UV-Vis spectra of feed toluene solution and solutions after filtration. (**b**) Filtration efficiency of PVA-UVCNP during a dead-end filtration experiment with initial toluene/xylene concentration of 40 mg/L and solution feed of $10 \, \text{L} \, \text{h}^{-1} \, \text{m}^{-2}$.

According to Cimadoro et al., when water permeates through a PVA electrospun membrane, the nanofibers undergo a swelling process, reducing the size of the pores and increasing tortuosity. This allows the retention of nanoparticles down to 20 nm in size [17]. Therefore, when contaminated water is filtered through the PVA-CNP membrane, the swelling of the nanofibers prevents the passage of the CNPs (of sizes between 45 and 105 nm) from side B to side A and their release into the filtered water. Figure S2 shows the FE-SEM micrograph of Side A after completing the toluene adsorption test, confirming that the CNPs remain attached to the PVA nanofibers on Side B of the membrane and do not pass to the other side.

At a 40 mg/L toluene concentration and solution feed of $10 \, \text{L} \, \text{h}^{-1} \, \text{m}^{-2}$, the filtration efficiency of the PVA-UVCNP membrane lay between 82% and 87% for three consecutive filtration cycles (see Figure 7b). This efficiency drops significantly in the 4th cycle to less than 70%. The removal efficiency decreases with the following filtration cycles down to 20% for the 7th cycle because the CNPs go near their maximum adsorption capacity. This way, the PVA-UVCNP membrane reached a high toluene adsorption capacity of $(647 \pm 20) \, \text{mg/g}_{\text{UVCNP}}$ after 35 mL of solution flow. A similar adsorption capacity of $(666 \pm 24) \, \text{mg/g}_{\text{UVCNP}}$ was obtained for xylene. The high adsorption capacity exhibited by PVA-UVCNP is promising when compared to those values of this parameter found in the literature for carbon sorbents [26,42].

The filtration of water and toluene altered Side B of the membrane. Particularly, toluene adsorption produced a relative increase of C=C stretching vibrations at 1625 cm$^{-1}$ and in the asymmetric bending of C-H groups at 1458 cm$^{-1}$, as shown in Figure 8. Adsorbed water led to an increase in the O-H vibration band. In contrast, filtration of toluene did not alter Side A of the membrane. Figure S3 shows that the IR spectrum of Side A after toluene filtration is identical to that of Side A after a control filtration using water without toluene. A control experiment without toluene was necessary, given that the wetting of the membrane alters its IR spectrum. Particularly, it noticeably increases the O-H vibration band at 3300 cm$^{-1}$ due to adsorbed water (see Figure S4).

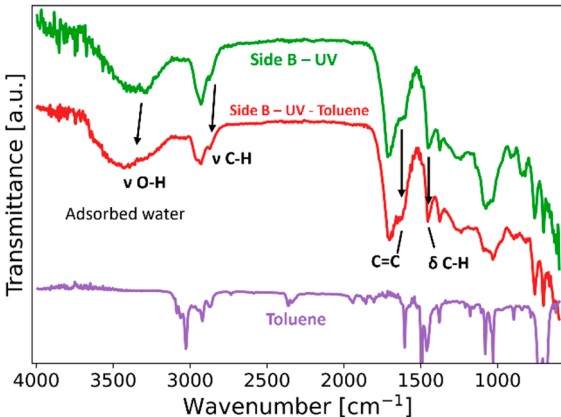

**Figure 8.** FTIR spectra of Side B after UV exposure and Side B after the toluene filtration experiment. The IR spectrum of toluene is also shown for reference.

To understand the selective sorption ability of the membrane, it is useful to study the interfacial energies between the materials involved. For toluene to be successfully absorbed by the membrane, it must have a higher affinity to the polymer than water. This means that the interfacial energy between toluene and the membrane material should be lower than that between toluene and water. Also, the membrane, or some part of it (in this case, the UVCNPs), should be more attracted to toluene than water.

The liquid-solid ($\gamma_{L\text{-}S}$) interfacial energy may be calculated by combining the equation used to obtain the solid surface tension, as shown by Ribba et al. [43], with Wenzel's equation. The result is shown in equation 3, where $\gamma_L$ is the surface tension of the liquid phase, $\theta$ is the contact angle, and $V_L$ and $V_S$ are the molar volumes of liquid and solid, respectively. The roughness factor $r$ is defined as the ratio between the actual surface area and the apparent surface area of a rough surface.

$$\gamma_{L-S} = \gamma_L \left[ \left(1 + \frac{cos(\theta)}{r}\right) \left(\frac{\left(V_L^{\frac{1}{3}} + V_S^{\frac{1}{3}}\right)^2}{4 V_L^{\frac{1}{3}} V_S^{\frac{1}{3}}}\right)^2 - \frac{cos(\theta)}{r} \right] \tag{3}$$

For this study, toluene (T), xylene (X), and water (W) were used, with their surface tension values being $\gamma_T$ = 27.9 mN/m, $\gamma_X$ = 29.6 mN/m, and $\gamma_W$ = 72 mN/m, respectively, as tabulated. The solid phase to consider is the UV-exposed CNPs, as it is the hydrocarbon-adsorbent part of the membrane. A range of values for $V_S$ was estimated by considering previous works [21,28]. The contact angle in equation 3 corresponds to the contact angles measured on Side B of the membrane after UV exposure ($\theta_T$ = 0°, $\theta_X$ = 0°, $\theta_W \lesssim$ 10°). Finally, a range of values for the roughness factor $r$ was also estimated, considering the morphology of the CNP structure over the fibers. Using all this data, limit values for the Toluene-UVCNP, Xylene-UVCNP, and Water-UVCNP interfacial energies were calculated, as shown in Table 1. Toluene-Water and Xylene-Water interfacial energies ($\gamma_{T\text{-}W}$, $\gamma_{X\text{-}W}$)

were measured in the laboratory, resulting in $(37 \pm 2)$ mN/m and $(38 \pm 1)$ mN/m (21 °C), respectively, which overlap with the values obtained in other studies [44].

**Table 1.** Energy values of the liquid-solid and liquid-liquid interfaces.

| Interfacial Energy ($\gamma$) | Water-UVCNP (mN/m) | Toluene-UVCNP (mN/m) | Xylene-UVCNP (mN/m) | Toluene-Water (mN/m) | Xylene-Water (mN/m) |
|---|---|---|---|---|---|
| Lower bound | 12.7 | 2.5 | 2.8 | $37 \pm 2$ | $38 \pm 1$ |
| Upper bound | 31.7 | 9.1 | 9.5 | | |

These results show that $\gamma_{\text{T-UVCNP}} < \gamma_{\text{T-W}}$, which means that toluene is energetically more compatible with the UVCNPs than it is with water. Also, $\gamma_{\text{T-UVCNP}} < \gamma_{\text{W-UVCNP}}$ implies that the UVCNPs have more affinity to toluene than water. These are the key factors for the amphiphilic membrane to be able to filter toluene dissolved in water. This analysis is completely analogous to xylene.

These interface energy relations also occur for the membrane with other hydrocarbons, such as benzene and oil, as can be estimated from the results of previous works [28,29]. In particular, the range of interfacial energies between cyclohexane and UVCNPs is one of the lowest: 2.3–8.0 mN/m, while the interfacial energy between cyclohexane and water is one of the highest (48 mN/m). These values present great potential for optimal cyclohexane interception. Considering the great sorption capacity achieved by the membrane, this technology has promising potential for the removal of BTEX and VOCs in general and even oil-in-water emulsions.

Given the availability of industrial-scale equipment for electrospinning production and RF plasma treatment, the process shown in this work is easily scalable and the PVA–UVCNP membrane produced is able to confine residual VOCs, preventing their release to the environment.

## 4. Conclusions

We have developed a simple, low-cost, and scalable process to produce a Janus membrane by combining the electrospinning of hydrophilic PVA and plasma deposited superhydrophobic NPs. SEM and FTIR studies showed that the differences in wettability properties on both sides of the PVA–CNP membranes arise from their morphologies and different chemical structures associated with the predominant materials on each side.

FTIR and wetting studies showed that UV exposure successfully transformed the CNPs superhydrophobic hierarchical structure with affinity to hydrocarbons into a hydrophilic one without losing the adsorption capacity towards organic contaminants. The UV-modified PVA–CNP membrane was able to selectively adsorb toluene and xylene dissolved in water due to the interfacial energies between the materials at play, showing high efficiency and high sorption capacity $((647 \pm 20)$ mg/g and $(666 \pm 24)$ mg/g, respectively). Moreover, the membrane could be used for three permeation cycles without significant efficiency loss.

The membrane shown in this work has great potential for water remediation, including the removal of BTEX from produced water as well as separating oil-in-water emulsions.

**Supplementary Materials:** The following supporting information can be downloaded at: https://www. mdpi.com/article/10.3390/colloids6040066/s1, Video S1: Wettability demonstration. Figure S1: UV-Vis spectra of the toluene solution before and after a filtration experiment through a control PVA membrane. Figure S2: FE-SEM image of the membrane after toluene adsorption. Figure S3: FTIR spectra of Side A of the membrane after filtration of toluene and a control solution using solely water. Figure S4: FTIR spectra of Side A of the membrane before and after filtration of water.

**Author Contributions:** Conceptualization, S.G., F.T. and D.G.; methodology, N.T.; formal analysis, N.T., P.G.-S. and F.T.; writing—original draft preparation, N.T., P.G.-S. and S.G.; writing—review and editing, N.T., P.G.-S., F.T., D.G. and SG; supervision, S.G. and D.G.; project administration, S.G.; funding acquisition, S.G. All authors have read and agreed to the published version of the manuscript.

**Funding:** This research was funded by the University of Buenos Aires-UBA (UBACYT 2018–2020 N° 200201701 00381BA), ANPCyT (PICT 2017-2362 and PICT STARTUP 2021-00017), and the National University of the West (UNO- 06-2020). MINCyT ("Programa Ciencia y Tecnología Contra el Hambre" IF-2021-4378615-APN-SSCI#MCT).

**Data Availability Statement:** The data presented in this study are available in the article.

**Conflicts of Interest:** The authors declare no conflict of interest.

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
