# Peer review of "Turning a Novel Janus Electrospun Mat into an Amphiphilic Membrane with High Aromatic Hydrocarbon Adsorption Capacity"

_colloids, doi:10.3390/colloids6040066_

Round 1

Reviewer 1 Report

This article reports a composite membrane of hydrophobic carbon nanoparticles synthesized by plasma polymerization on a PVA hydrophilic electrospun membrane, and the hydrophilicity of CNPs is enhanced by UV irradiation. The nanostructured amphiphilic membrane capable of rapidly removing hydrocarbons present in water under the solubility limit, by letting the water pass through while retaining the contaminant. This is a throughout study. It could be acceptable after clarifying the following issues.

1. Could the authors explain how the hydrophobic CNPs synthesized by plasma polymerization are bound to the hydrophilic electrospun PVA? Does UV treatment have any effect on their binding effect?

2. Could the authors provide the morphology of the CNPs on the Side B after UV irradiation?

3. Could the authors provide XPS analysis of CNPs on the membrane after toluene adsorption to explore the adsorption mechanism?

4. The authors performed analytical tests on toluene adsorption and compared it to water by liquid-solid energy. Could the authors add several other organic compounds and verify the optimal interception of contaminants based on solubility parameters?

5. The authors mentioned that the swelling of the PVA might prevent the liberation of CNPs. Could the authors provide the morphology of the composite membrane after multiple pollutant interceptions?

Round 2

Reviewer 2 Report

The response addressed the problems well and improved the quality of the manuscript.